# Polariton condensation into vortex states in the synthetic magnetic field of a strained honeycomb lattice

Cristóbal Lledó[1,2]⋆, Iacopo Carusotto[3]† and Marzena H. Szymańska[1]‡

**1** Department of Physics and Astronomy, University College London,
Gower Street, London, WC1E 6BT, United Kingdom
**2** Institut quantique & Département de Physique, Université de Sherbrooke,
Sherbrooke, Quebec J1K 2R1, Canada
**3** INO-CNR BEC Center and Dipartimento di Fisica, Università di Trento,
I-38123 Povo, Italy

⋆ cristobal.lledov@gmail.com, † iacopo.carusotto@unitn.it, ‡ m.szymanska@ucl.ac.uk

## Abstract

Photonic materials are a rapidly growing platform for studying condensed matter physics with light, where the exquisite control capability is allowing us to learn about the relation between microscopic dynamics and macroscopic properties. One of the most interesting aspects of condensed matter is the interplay between interactions and the effect of an external magnetic field or rotation, responsible for a plethora of rich phenomena—Hall physics and quantized vortex arrays. At first sight, however, these effects for photons seem vetoed: they do not interact with each other and they are immune to magnetic fields and rotations. Yet in specially devised structures these effects can be engineered. Here, we propose the use of a synthetic magnetic field induced by strain in a honeycomb lattice of resonators to create a non-equilibrium Bose-Einstein condensate of light-matter particles (polaritons) in a rotating state, without the actual need for external rotation nor reciprocity-breaking elements. We show that thanks to the competition between interactions, dissipation and a suitably designed incoherent pump, the condensate spontaneously becomes chiral by selecting a single Dirac valley of the honeycomb lattice, occupying the lowest Landau level and forming a vortex array. Our results offer a new platform where to study the exciting physics of arrays of quantized vortices with light and pave the way to explore the transition from a vortex-dominated phase to the photonic analogue of the fractional quantum Hall regime.



# 1   Introduction

Many-body systems in the presence of synthetic magnetic fields offer the exciting opportunity of finding new topological phases of matter with fractional excitations [1]—a holy-grail of condensed-matter physics and the pillar of topological quantum computing [2,3]. Beyond ultracold atomic gasses [4–7], photonic systems seem to be an ideal platform for this purpose thanks to the exquisite capabilities state preparation and detection as well as direct access to correlations [8–14]. However, to achieve these exotic phases of light a challenging combination of strong interactions and synthetic magnetic fields is required. So far, only partial results have been obtained. On the one hand, strong effective photon-photon interactions are already routinely attained in a variety of photonic platforms including superconducting quantum circuits for microwave photons [15–17] or, in the optical regime, Rydberg polaritons [18], and are underway in exciton-polaritons [19–22]. On the other hand, considerable effort has been devoted to engineering synthetic magnetic fields for light by various approaches [13,23]; for example, by including magnetic materials in microwave- and telecom-frequency cavities [24–26], by Floquet engineering [27] or by carefully designing non-trivial lattice geometries [28–31]. Yet, even though there has been one successful attempt in creating few-body photonic Laughlin states [32], combining interactions with synthetic fields in a scalable platform for many-photon states is still needed.

In this work we propose a method to stabilize a non-equilibrium Bose-Einstein condensate of polaritons into a rotating state generated by a synthetic magnetic field. We make use of the recently developed platform of strained honeycomb lattices. Based on the idea of a strained graphene layer [33, 34], a lattice of photonic resonators can be carefully designed to have a spatial gradient in its hopping strengths. The synthetic strain induces a synthetic magnetic field which has opposite sign in each of the two Dirac valleys of the honeycomb lattice, leading to the formation of Landau levels in the middle of the energy spectrum [35, 36] (see Fig.1). The first evidence of Landau level quantization in a strained honeycomb lattice for photons was obtained in coupled dielectric optical waveguides [29], and more recently, the photonic $n = 0$ Landau level wavefunction was directly imaged in a lattice of macroscopic microwave resonators [30] and of coupled semiconductor micropillar cavities [31].

We theoretically anticipate that, thanks to the competition between drive and decay, a robust polariton condensate can be stabilized in the the $n = 0$ Landau level using an incoherent pump which selects a single sublattice. The properties of the steady state are determined by the interplay between interactions and the synthetic magnetic field. The Dirac valley symmetry is spontaneously broken via the nonlinearity which comes from polariton interactions and—in a smaller proportion—from the incoherent pump saturation. In this way, just one of the two

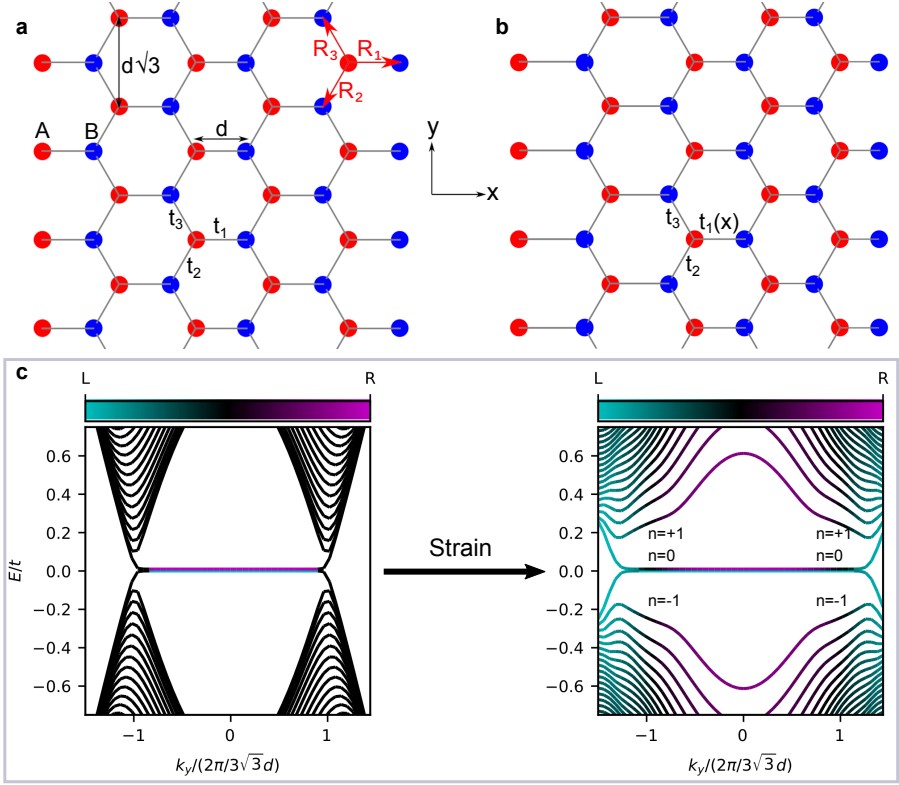

Figure 1: **Honeycomb lattice and single-particle spectrum. a, b** Sketch of the un-strained (left) and strained (right) honeycomb lattices. We use bearded terminations in the $x$ direction and armchair ones in the $y$ direction. A generic strain is encoded in the $x$-dependent hopping strength along the horizontal links $t_1(x) = t(1+\tau x/3d)$ and is graphically indicated as a gradient in the link's length in **b**. Hopping along the other links is constant $t_{2,3} = t$ and $\tau$ is the strain magnitude. **c** The single-particle 1D energy bands of the unstrained and strained lattices with periodic boundary conditions along $y$. The color indicates the mean $x$-position of the eigenstates, $\langle \phi | \hat{x} | \phi \rangle$. Cyan corresponds to the (L)eft edge of the lattice and magenta to the (R)ight. The Landau levels $n = 0, \pm 1$ are labeled in the spectrum of the strained lattice.

possible orientations of the synthetic field is selected, and thus, the condensate forms a vortex lattice analogous to that observed in a rapidly rotating ultracold gas [4, 37–39] or a type-II superconductor in the presence of an external magnetic field [40–42].

Our proposal naturally applies to a variety of photonic platforms. For the sake of concreteness, we will focus our discussion on the specific case of a honeycomb lattice of semiconductor micropillar cavities for exciton-polaritons, as used in the recent experiments [31, 43–46] and theoretical works [35, 47, 48] to then comment what are the challenges and advantages of other platforms in view of our proposal.

## 2 Results

### 2.1 Strained honeycomb lattice

The honeycomb lattice is made of two sublattices $A$ and $B$. We model the kinetic energy of polaritons using a tight-binding Hamiltonian with nearest-neighbor hopping between the two

sublattices,

$$H = -\sum_{\mathbf{r},j} t_j(\mathbf{r})\hat{a}^\dagger(\mathbf{r}-\mathbf{R}_j)\hat{b}(\mathbf{r}) + \text{h.c.}, \tag{1}$$

where $\hat{a}$ and $\hat{b}$ annihilate a polariton in the $A$ and $B$ sublattices, respectively, and h.c. represents the Hermitian conjugate. The vector $\mathbf{r} = (x, y)$ indicates real-space coordinates and it labels all the positions of the $B$ sites, while the vectors $\mathbf{R}_j$, depicted in Fig. 1a, connect different sites. The hopping energies $t_j(\mathbf{r})$ correspond to the links associated with the vectors $\mathbf{R}_j$. We consider a hopping gradient in the horizontal links along $x$, choosing $t_1(x) = t(1 + \tau x/3d)$, where $d$ is the lattice spacing, and $t_2 = t_3 = t$. The constant $\tau \geq 0$ determines the degree of strain in the lattice and, for a given size of the lattice in the $x$ direction, it has a maximum value beyond which $t_1(x)$ undergoes a sign change for negative positions $x$ (this is a Lifshitz transition and we avoid it).

For momenta near the two nonequivalent Dirac points,

$$\mathbf{K}_\xi = (K_{\xi,x}, K_{\xi,y}) = \xi\left(\frac{2\pi}{3d}, \frac{2\pi}{3\sqrt{3}d}\right), \tag{2}$$

where $\xi = \pm 1$, the strain gives rise to a pseudomagnetic field $e\mathbf{B} = \xi 2\hbar\tau/9d^2\hat{z}$ (see Appen. A or [35]), which has opposite orientation in each Dirac valley. The effect that this has over the single-particle energy bands can be seen in Fig. 1c (note that, due to the strain, the only good quantum number is $k_y$). There are (tilted) Landau levels labeled by $n = \pm 1, \pm 2, \dots$. To a very good approximation, their energies are given by $\varepsilon_n = \text{sgn}(n)t\sqrt{\tau|n|(1-\xi q_y d)}$, where $q_y = k_y - K_{\xi,y}$ is the (small) momentum-space distance measured from the Dirac point. For our purposes, the only important information about their wavefunctions is that they equally occupy both sublattices, $A$ and $B$, for any non-zero $n$. On the other hand, the band of $n = 0$ Landau levels have zero dispersion ($\varepsilon_0 = 0$) and, most importantly, are completely localized in the $B$ sublattice (it would be in the $A$ sublattice if $\tau$ was negative). Their wavefunctions read

$$\phi_{n=0,B}(x, q_y) \propto e^{iK_{\xi,x}x}e^{-(x+\xi\ell_B^2 q_y)^2/2\ell_B^2}, \tag{3}$$

with $q_y = k_y - K_{\xi,y}$, and $\phi_{n=0,A} = 0$. This sublattice polarized wavefunctions can be recognized (besides the $x$-dependent phase) as the lowest Landau level states in the Landau gauge [49], given by a plane wave in the periodic $y$-direction and a Gaussian in $x$, with a guiding center $x_0 = -\xi\ell_B^2 q_y$ determined by the $q_y$ wavevector and a width determined by the magnetic length $\ell_B = \sqrt{\hbar/e|B|} = 3d/\sqrt{2\tau}$. This band of lowest Landau level states is the only family of bulk states in the system with sublattice polarization. We will take advantage of this to induce polariton condensation into the $n = 0$ Landau level [50]: given the full sublattice polarization of this state, we expect that the mode selection mechanism based on spatial overlap with the pumped sublattice will largely exceed other mechanisms that may be active in polariton systems, such as energy relaxation, the exciton-photon fraction of modes, and the amount of localization of the wave function at the pillars.

To better simulate a real experimental implementation, in the following we always consider large but finite lattices of $N_x$ links along $x$ and $N_y$ unit cells along $y$. As a reference, the diagram in Fig 1a shows a lattice with $N_x = 5$ and $N_y = 4$.

## 2.2 Incoherent pump, decay and interactions

To model the Bose-Einstein condensation into the $n = 0$ Landau level we have performed numerical simulations of interacting polaritons with binary interactions of strength $U$ in a finite strained honeycomb lattice, under the action of an incoherent pump and polariton losses

(dominated by photon leakage out of the microcavities). The system is well described by stochastic Gross-Pitaevskii equations [51] that read

$$i\hbar\partial_t\psi_A = \text{Hop}(\psi_A \to \psi_B) + U|\psi_A|^2_-\psi_A + \frac{i\hbar}{2}\left(\frac{P_A}{1+|\psi_A|^2_-/n_s} - \gamma\right)\psi_A + iW_A, \qquad (4)$$

$$i\hbar\partial_t\psi_B = \text{Hop}(\psi_B \to \psi_A) + U|\psi_B|^2_-\psi_B + \frac{i\hbar}{2}\left(\frac{P_B}{1+|\psi_B|^2_-/n_s} - \gamma\right)\psi_B + iW_B. \qquad (5)$$

For simplicity, we have omitted the spatio-temporal dependence of the macroscopic wavefunction $\psi = \psi(\mathbf{r}, t)$, of the pump profile $P = P(\mathbf{r})$, and of the zero-mean complex Gaussian noise with $\left\langle W_X(\mathbf{r}', t')W_Y^*(\mathbf{r}, t)\right\rangle = \delta_{XY}\delta_{\mathbf{r}'\mathbf{r}}\delta(t - t')(\hbar^2/2)(P_X/(1 + |\psi_X|^2_-/n_s) + \gamma)$, where the average is over stochastic realizations. The Hop terms correspond to the hopping between nearest neighbors and depend explicitly on the strain (see Appen. B for the explicit expression). $U$ is the polariton interaction strength, $n_s$ the saturation density, and $\gamma$ the decay rate. Polaritons in micropillars have an on-site interaction energy smaller than the linewidth, $U/\hbar\gamma < 1$; in our results below we use modest ratios perfectly within the reach of current experiments (see e.g. Ref. [52]). Underlying these equations is the assumption that there is a high-energy excitonic reservoir, excited by an external pump, which leads to the saturable incoherent pump[1] in Eqs. (4) and (5). These equations are obtained from a truncated Wigner approximation and thus the Wigner commutator is subtracted from the intensity, $|\psi|^2_- = |\psi|^2 - 1$.

If the occupation of the excitonic reservoir is substantial, there is an additional polariton energy blueshift—not included in Eqs. (4) and (5)—which is induced by the reservoir formed in the pumped region. Since we will consider a modulated pump acting only on some sites and not on all of them, this extra term could distort the single-particle energy bands. We later discuss the possible complications that might arise as well as possible workarounds.

The results that we are going to show in the following of the work refer to the long-time limit of the system evolution. A study of the additional features that may occur during the switch-on transient, such as Kibble-Zurek vortex nucleation phenomena [48], go beyond the scope of this work.

## 2.3 Condensation into the $n = 0$ Landau level

In order to fully exploit the sublattice polarization of the $n = 0$ Landau level, we start by considering a spatially modulated incoherent pump acting only on the $B$ sublattice ($P_A = 0$). When the pump overcomes losses, $P_B \gtrsim \gamma$, polaritons condense in the $n = 0$ Landau level, since this is the only single-particle state with this sublattice polarization. Depending on how many sites the incoherent pump covers, we find a BEC with zero, one, or many vortices.

We first discuss the situation where the size of the pumped region is not too large compared to the magnetic length $\ell_B$. In Fig. 2a,b we show the stationary-state intensity, phase, and spectral density for two Gaussian pumps with the same maximum intensity but different widths, namely $P_B(r)/\gamma = 1.3e^{-r^2/2\sigma^2}$, with $\sigma = 2\ell_B, 5\ell_B$, respectively. In the former case, the pump region is not wide enough to accommodate a vortex, while in the latter, a clear anti-vortex ($-2\pi$ phase circulation) can be seen in the center of the lattice. In the plotted spectral densities $\left\langle|\psi(\omega, k_y)|^2\right\rangle$, one can see that the condensate occupies a single Dirac valley predominantly. Which of the two valleys is chosen for condensation is randomly chosen

---

[1]This form of pump as well as its influence in the noise correlations come from the incoherent process described by a Lindblad dissipator $C_X(\hat{\psi}_X^\dagger\hat{\rho}(t)\hat{\psi}_X - \frac{1}{2}\{\hat{\psi}_X\hat{\psi}_X^\dagger, \hat{\rho}(t)\})$, where the gain coefficient $C_X$ depends on the exciton-reservoir density. In the Wigner representation and after adiabatically eliminating this reservoir under the assumption that it instantaneously follows the low-energy polariton density, the coefficient becomes $C_X = P_X/(1 + |\psi_X|^2_-/n_s)$.

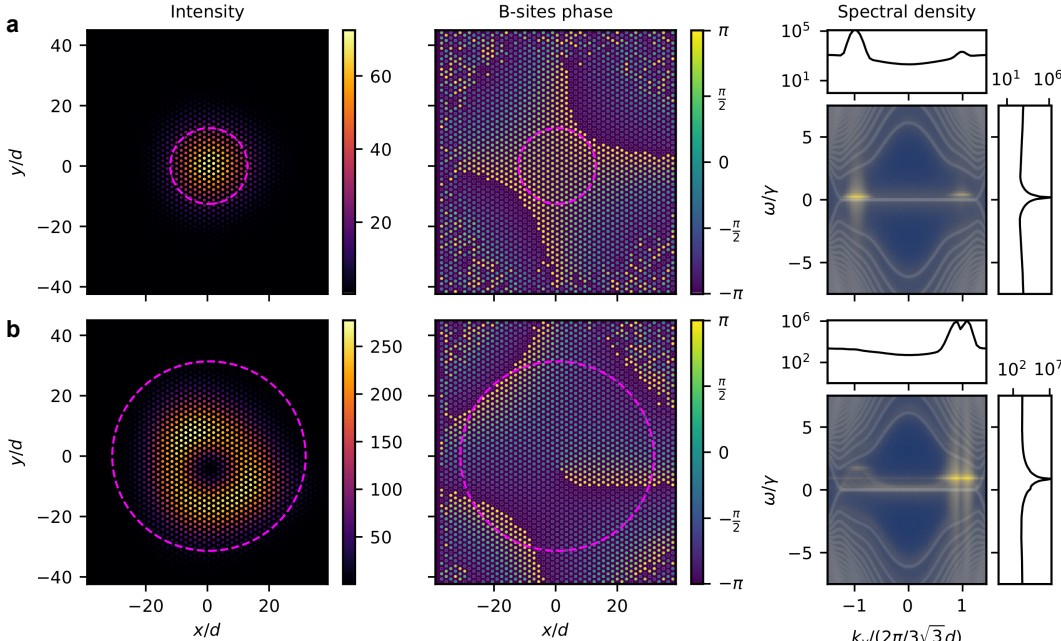

Figure 2: **BEC in the n = 0 Landau level.** Intensity (left) and phase (middle) spatial profiles, and spectral density $\left\langle |\psi(\omega, k_y)|^2 \right\rangle$ (right) for a Gaussian pump with maximum $1.3\gamma$ and width $\sigma = 2\ell_B$ in **a** and $\sigma = 5\ell_B$ in **b**. The interior of the magenta (dashed) circles indicates the region where the pump is larger than the losses. In the phase profiles, only the $B$ sublattice is shown, and for clarity we have removed the phase-shift from to the Dirac momentum by plotting the phase of $\langle \exp(-i\mathbf{K}_\xi \cdot \mathbf{r})\psi_B(\mathbf{r})\rangle$, where $\mathbf{K}_\xi = \xi(2\pi/3d, 2\pi/3\sqrt{3}d)$, with $\xi = \pm 1$. In **b** a clear anti-vortex ($-2\pi$ circulation) is displayed. In the right column we have superimposed the spectral density to the single-particle spectrum, shown in dim white lines. The parameters are $J/\hbar\gamma = 10$, $n_s = 10^3$, $U/\hbar\gamma = 0.005$, $\tau = 0.06$, $N_x = 51$, and $N_y = 50$. We have averaged over 100 stochastic realizations.

at each instance of the condensation process and depends on the randomly chosen initial conditions and on the noise $W_{A,B}$, if any. The vortex has its origin in the single-particle Landau states, so (besides the energy blueshift seen in the spectral density) interactions play no role on the vortex formation at these pumped-region sizes. Indeed, the (normalized) stationary state analyzed in Fig. 2b has a remarkable overlap of $\left| \sum_{\mathbf{r}} \phi^*(\mathbf{r}) \cdot \psi_B(\mathbf{r})/\mathcal{N} \right| \approx \%68$ (%99 intensity match) with the slightly spatially shifted $n = 0, m = -1$ symmetric-gauge Landau state $\phi(x, y) \propto (x - iy)e^{-(x^2+y^2)/4\ell_B^2}$ with angular momentum $m\hbar = -\hbar$ [49]. As usual, this symmetric-gauge Landau state can be written as a linear combination of Landau-gauge Landau states of different $k_y$'s.

On the contrary, interactions play instead a key role in the case of wider incoherent pump profiles, where they make condensation in a single Dirac valley—as opposed to in both valleys simultaneously—more likely. In standard BECs at thermal equilibrium, the interaction energy is minimized by choosing among the single-particle ground states the one which has the most homogenous density in real space. Here, a similar mechanism hinders the condensate from occupying both valleys at the same time, which would entail large density inhomogeneities due to destructive interference of states with opposite angular momenta. This is important for forming a vortex array, as the two opposite pseudomagnetic fields at the two Dirac valleys compete and suppress vortices. In Fig. 3, we show late-time snapshots of the real-space

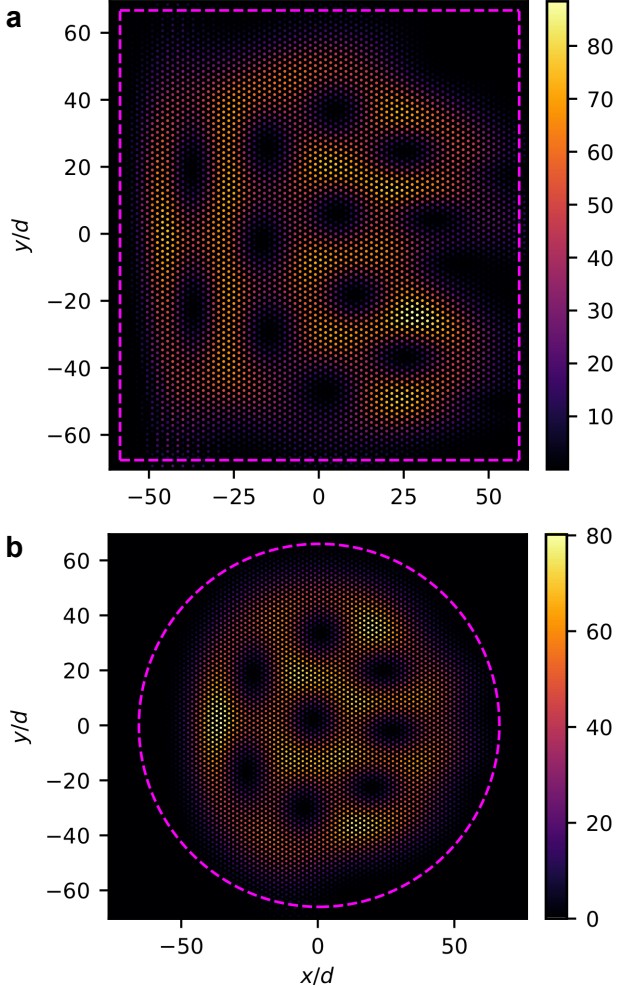

Figure 3: **Vortex arrays for large lattices and wide top-hat pumps**. Real-space intensity for a rectangular (**a**) and circular (**b**) top-hat pump. These are late-time snapshots of mean-field trajectories ($W_{A,B} = 0$). The vortex array rotates in the sense determined by the orientation of the chosen pseudomagnetic field. All $B$ sites inside the magenta dashed rectangle or circle are pumped with the same intensity. The parameters are $J/\hbar\gamma = 10$, $n_s = 10^3$, $P_B/\gamma = 1.05$, and $N_y = 80$ in both cases, and $U/\hbar\gamma = (4,5) \times 10^{-3}$, $\tau = (0.045, 0.039)$, and $N_x = (81, 101)$ in (**a**, **b**).

intensity profile for large rectangular and circular top-hat incoherent pumps (drawn on the images). In both cases, condensation occurs in a single Dirac valley and several vortices form. Note that for such wide pump profiles, the condensate does not reach a stationary state but keeps evolving in time even at the mean-field level ($W_{A,B} = 0$). In all cases, the vortex lattice tends to rotate in the sense determined by the sign of the pseudomagnetic field corresponding to the chosen Dirac valley. For a non-circular pump profile, the shape of the vortex lattice and its motion are no longer regular, but displays chaotic trajectories. If noise is added ($W_{A,B} \neq 0$), in each trajectory the vortices diffuse in a slightly different way. Thus, with time, the vortices will be smeared out in the stochastic average. Nevertheless, this process is slow ($\sim 100\gamma^{-1}$), so vortex lattices should be observable in a single shot experiment where the intensity profile is obtained as a time average of a single realization. Two videos showing the dynamics corresponding to the the snapshots of Figs. 3a and b can be found in [53].

If interactions are switched off ($U = 0$), the saturation nonlinearity in Eq. (5)—which

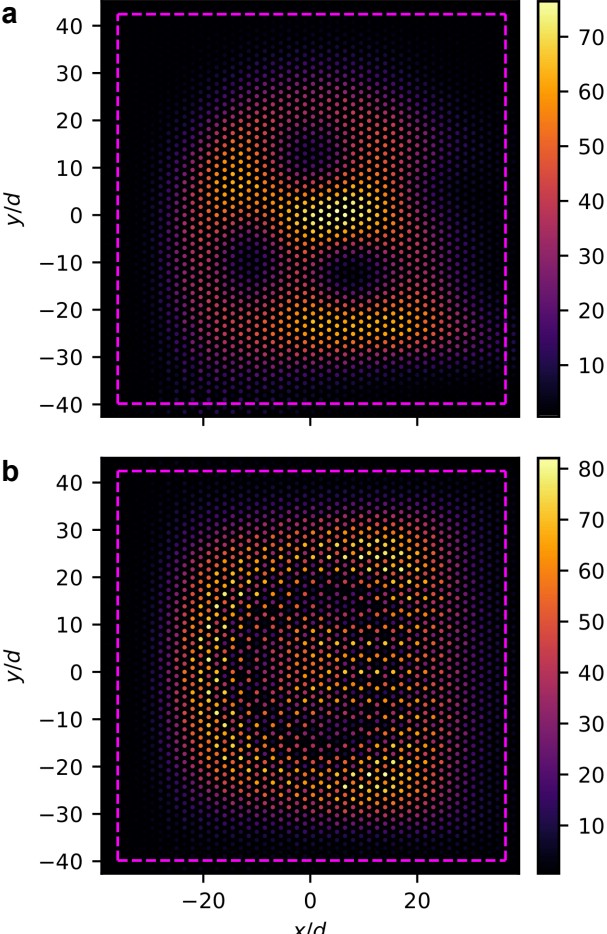

Figure 4: **Comparison of condensation with and without interactions.** Steady-state real-space intensity profile for $U/\hbar\gamma = 0.005$ (**a**) and $U = 0$ (**b**). Condensation takes place in one Dirac valley in **a**, while in both valleys in **b**. All $B$ sites inside the magenta dashed rectangle are pumped with the same intensity. The rest of the parameters are $J/\hbar\gamma = 10$, $n_s = 10^3$, $P_B/\gamma = 1.05$, $\tau = 0.06$, $N_x = 51$, and $N_y = 50$. We have averaged over 100 stochastic realizations.

plays a similar role to polariton-polariton interactions—is not always strong enough to efficiently prevent condensation in both Dirac valleys. To illustrate this point, let us compare the interacting and non-interacting cases in a moderate-sized lattice with a broad rectangular top-hat pump. In Fig. 4a, for $U \neq 0$, three anti-vortices are found as the polaritons have condensed in the positive-momentum Dirac valley. In Fig. 4b we show, instead, the non-interacting $U = 0$ case where both Dirac valleys are occupied, such that no vortices are formed. Instead, we obtain a spatially oscillating density pattern formed from overlapping condensates at the two valleys. For large pump profiles, it is rare but not impossible, even for sizable $U$, to end up in configurations of two almost spatially separated condensates in different valleys with an overlapping boundary region. An example of such situation is shown in Fig. 5, where there is a single anti-vortex in the bottom and three vortices in the middle-top. However, unless $U = 0$ the condensates in the two valley are spatially separated and their overlap is hardly as great as in Fig. 4(b).

The fact that the two opposite pseudomagnetic fields can suppress the formation of vortices posses the question of how many vortices are typically found in a given realization of the

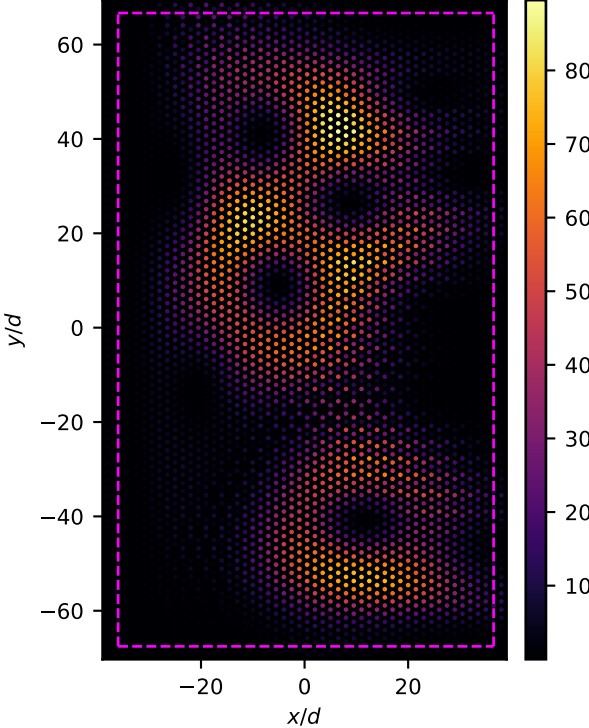

Figure 5: **Spatially-separated condensates in different Dirac valleys.** Late-time snapshot of the real-space intensity in a mean-field trajectory ($W_{A,B} = 0$). There is one anti-vortex in the bottom, three vortices in the middle-top, and an interference pattern in between. All $B$ sites inside the magenta dashed rectangle are pumped with the same intensity. Same parameters as in Fig. 4a but with $N_y = 80$.

condensation process. If we had a real magnetic field and the system was at thermal equilibrium, the number of vortices would be determined by the ratio between the total magnetic flux piercing the system and the magnetic-flux quantum, so would thus be proportional to the system size. In our case, the number of vortices found fluctuates from one realization to another. We statistically analyze 100 realizations of the mean-field dynamics with different random initial conditions to look at how the number of vortices depend upon the number of unit cells $N_y$ in the $y$-direction. We consider a top-hat pump over all $B$-sites but the outermost (as in in Fig. 4). In Fig. 6 we show the statistics of the total number of vortices (and/or anti-vortices, irrespective of the topological charge) for different values of $N_y$. We compare the interacting and non-interacting cases. We can see that for $U = 0$ the number of vortices grows much slower as $N_y$ is increased than for the finite $U$ case. As we have discussed earlier, this is because interactions favor the selection of a single Dirac valley for condensation—or, in other words, one of the two orientations of the pseudomagnetic field—and thus there is no competition for vortex formation.

## 2.4 Robustness to disorder and the effect of reservoir-induced blueshift

In order to verify the actual feasibility of our proposal, we have checked that the BEC in the $n = 0$ Landau level is robust against those complications that are most likely to occur in concrete experiments. First of all, we have verified that our predictions remain valid if some sizable fraction of the incoherent pump sneaks into the $A$ sublattice, which is the one not occupied by the $n = 0$ states. It is possible—for any lattice size—to have $P_A \sim 0.5 P_B$ without any

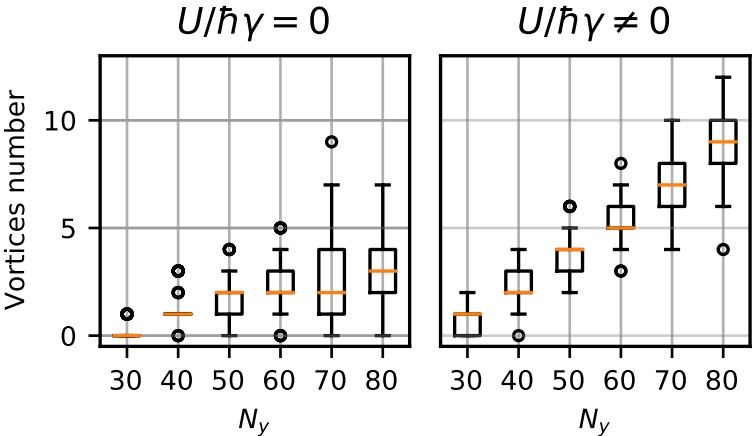

Figure 6: **Boxplot for the vortex-number statistics** of 100 mean-field (zero noise $W_{A,B} = 0$) realizations of different random initial conditions $\psi(x_i, y_j, t = 0)$, for $N_y = 30, 40, \ldots, 80$, with (right) and without (left) interactions. For each data set, the lower bar is at the lowest datum above $Q_1 - 1.5(Q_3 - Q_1)$, and the upper bar at the highest datum below $Q_3 + 1.5(Q_3 - Q_1)$, where $Q_1$ and $Q_3$ are the first and third quartiles. The median is shown with the orange line, the box encloses data between $Q_1$ and $Q_3$, and the outliers are indicated by circles. We considered a top-hat pump over all $B$-sites but the outermost (as in in Fig. 4). The parameters are $J/\hbar\gamma = 10$, $P_B/\gamma = 1.1$, $\tau = 0.06$, $N_x = 51$, and $U/\hbar\gamma = 0.005$ in the case finite interactions.

qualitative change. The maximum amount of pump in the $A$ sublattice that the vortex configuration can tolerate depends on the system size as well as the pump intensity. For instance, for small lattices (e.g., $N_x = 11, N_y = 20$), both sublattices can be pumped with almost the same intensity near the BEC threshold. In general, it is convenient to avoid pumping the left-hand edge of the lattice where the propagating edge states (associated to the $n = 0$ Landau level) are found. If that region is strongly pumped, it can lead to chaotic behavior where many energy bands are occupied.

As previously pointed out, if the excitonic reservoir is substantially populated it leads to an on-site energy renormalization (blueshift) for the polaritons in the micropillars. Since we pump only the $B$ sublattice, the blueshift would affect just the pumped micropillars and would distort the Landau energy levels. Moreover, using a small Gaussian pump spot can lead to localization into the highest energy bands [54], with the energy blueshift playing a similar role to on-site energy disorder. We have checked that the vortex configuration is robust to the reservoir-induced blueshift as well as to on-site energy disorder, as long as they are smaller than $\hbar\gamma$.

In order to remove this possibly stringent constraint on the reservoir-induced blueshift, one can work in the weak light-matter coupling regime where the blueshift should be saturated and then it does not matter that some micropillars are pumped and others are not. In this case, the polariton would be predominantly photonic and the interactions very weak ($U \sim 0$), but vortices and vortex arrays can still form, just less efficiently as shown in Fig. 6. A different workaround for the energy blueshift is to tailor the polariton losses such that the loss in $A$ sublattice is larger than that of the $B$ sublattice. We have checked that when the decay rate is %15 larger in the $A$ sublattice, one can uniformly pump (slightly above BEC threshold) all the sites in the system and the $n = 0$ Landau level is still selected for condensation. In semiconductor micropillar cavities this could be achieved by modifying the Bragg mirrors of the $A$-sublattice's micropillars to increase their radiative decay. Note however that all complications arising from

the excitonic reservoir are exclusive to exciton-polaritons and should not arise in alternative platforms that we suggest in the following section.

# 3 Discussion

In addition to suggesting a new experimental platform where to study the many-body physics of vortex lattices in a novel non-equilibrium context, our work raises a number of further conceptual questions. The randomness in the Dirac valley selection poses the intriguing question whether it could be possible to bias the selection towards one valley to systematically form vortices or anti-vortices. One possibility would be to have a biasing coherent pump on one Dirac valley, which is slowly switched off in time in such a way that the condensate remains in the biased valley. This is a standard procedure in the study of symmetry-breaking phase transitions. Another exciting option to explore is to deform the lattice in a way that a single Dirac valley is dynamically favored and time-reversal symmetry gets effectively broken without the need for reciprocity-breaking elements, but with the help of the nonlinear mechanisms involved in condensation, as recently studied in simpler geometries [55].

In this work we focus on polariton micropillar lattices, but there are other promising potential platforms where to observe the predicted spontaneous formation of vortex arrays, including the evanescently-coupled optical waveguides of the experiments in refs. [27, 29] and circuit QED lattices [14, 15, 17, 56]. In the coupled optical waveguides experiments, the interplay of optical nonlinearities with topology has recently attracted a great attention [57, 58] and different losses for the two strained-honeycomb's sublattices have been used in ref. [59] to mimic the effect of an incoherent pump and study a non-Hermitian $\mathcal{PT}$-symmetry-breaking phase transition. Regarding the circuit QED platform, a main challenge would be the disorder of the hopping strength and the resonator energy, but there is great flexibility in lattice design [56] and the platform offers the possibility of having stronger interactions [15, 17] than those achievable in exciton-polaritons.

When applied to strongly-interacting photonic lattices ($U > \hbar\gamma$), our findings pave the way to studying the transition from the vortex-dominated BEC regime to the strong-correlations regime of the fractional quantum Hall physics. That transition has been first anticipated for ultracold atoms in a rotating trap, but so far its observation has remained experimentally challenging [4, 39]. It is in fact experimentally very hard to achieve high angular velocities and low atomic densities such that a fractional occupation of the lowest Landau level is achieved. While active work is being devoted to reach strong interactions for exciton-polaritons in semiconductors [19–22], first reports of experimental studies of the interplay of strong synthetic magnetic fields and strong interactions in photonic systems have recently started appearing in the literature using Rydberg polaritons in atomic media [32] or circuit-QED devices [60] giving concrete hopes for the success of this very ambitious scientific adventure.

## Data and Code Availability

All the numerical data and codes used in this article are available upon reasonable request to the first author.

## Acknowledgements

We thank A. Amo, J. Bloch, O. Jamadi, and S. Ravets for helpful discussions, and C. Mc Keever for comments on the manuscript. C.L. gratefully acknowledges the financial support of ANID through Becas Chile 2017, Contract No. 72180352. IC acknowledges support from the European Union Horizon 2020 research and innovation programme under grant agreement No. 820392 (PhoQuS) and from the Provincia Autonoma di Trento. M.H.S. gratefully acknowledges financial support from the Quantera ERA-NET cofund project InterPol (through the EPSRC Grant No. EP/R04399X/1) and EPSRC Grant No. EP/S019669/1.

## A  Analytical derivation of Landau levels

The analytical derivation of the Landau energies and states in a strained honeycomb lattice is well understood in the context of graphene [33,34]. Here we reproduce the results from [35, 36], which deal with a synthetic strain, as in our work.

The Landau level states and energies can be obtained by writing the single-particle bulk Hamiltonian in momentum space, $H(\mathbf{k}) = \mathbf{h}(\mathbf{k}) \cdot \boldsymbol{\sigma}$, where $\boldsymbol{\sigma} = (\sigma_x, \sigma_y)$ are Pauli matrices and $\mathbf{h}(\mathbf{k}) = (\text{Re } h(\mathbf{k}), -\text{Im } h(\mathbf{k}))$, with $h(\mathbf{k}) = -\sum_j t_j \exp(i\mathbf{k} \cdot \mathbf{R}_j)$. Due to the strain and so the lack of translational symmetry, we must consider a finite lattice along $x$ and thus $k_x$ is no longer a good quantum number. Moreover, $h(\mathbf{k})$ depends explicitly on $x$ via $t_1(x)$. Yet, to a very good approximation, the bulk Hamiltonian describes well the bulk physics [31,35,36]. This is because, as will be clear in the following, there is a wide range of parameters where orbits are well localized in the bulk in a region where the hopping $t_1$ does not change significantly. Having stated the necessary precautions, we proceed by expanding $h(\mathbf{k})$ for small momenta $\mathbf{q}$ near one of the two opposite Dirac points, $\mathbf{k} = \mathbf{K}_\xi + \mathbf{q}$, where $\xi = \pm 1$ and $\mathbf{K}_+ = (\frac{2\pi}{3d}, \frac{2\pi}{3\sqrt{3}d}) = -\mathbf{K}_-$, yielding

$$h(\mathbf{K}_\xi + \mathbf{q}) \approx v_D^x \hbar q_x - i v_D^y \xi (\hbar q_y + eA_y) \tag{6}$$

to first order in $x$ and $q$, where $v_D = (3dt/2\hbar)e^{\xi i 2\pi/3}$ is a Dirac velocity and $eA_y = 2\hbar\tau\xi x/9d^2$ is the synthetic vector field, giving rise to the two opposite magnetic fields $e\mathbf{B} = \nabla \times e\mathbf{A} = \xi 2\hbar\tau/9d^2 \hat{z}$.

To find the spectrum of $H(\mathbf{K}_\xi + \mathbf{q})$, we promote $h$ in Eq. (6) to an operator such that it becomes proportional to the annihilation operator of a displaced harmonic oscillator,

$$\hat{h} \approx \hbar t\sqrt{\tau}\hat{c} = v_D \hbar \hat{q}_x - i\frac{v_D}{\ell_B^2}(\hat{x} + \xi\ell_B^2 q_y), \tag{7}$$

where $q_y$ is the conserved $y$-momentum, $\ell_B = 3d/\sqrt{2\tau}$ is the magnetic length, and $[\hat{c}, \hat{c}^\dagger] = 1$. This suggests the eigenvectors of $H(\mathbf{K}_\xi + \mathbf{q})$ should be vectors of two components related to the solutions of a displaced harmonic oscillator, where the displacement is proportional to $q_y$, just as Landau-level states in the Landau gauge [49]. This is indeed the case. The eigenvalue problem $H(\mathbf{K}_\xi + \mathbf{q})\vec{\phi} = \varepsilon\vec{\phi}$ is

$$\hbar t\sqrt{\tau}\begin{pmatrix} 0 & \hat{c} \\ \hat{c}^\dagger & 0 \end{pmatrix}\begin{pmatrix} \phi_A \\ \phi_B \end{pmatrix} = \varepsilon\begin{pmatrix} \phi_A \\ \phi_B \end{pmatrix}, \tag{8}$$

and its solution consists of the $\varepsilon_{n=0} = 0$ energy and its associated eigenstate in Eq. (3), as well as higher and lower Landau energy levels labeled by $n = \pm 1, \pm 2, \dots$, with energy $\varepsilon_n = \text{sgn}(n)t\sqrt{\tau|n|}$ and eigenstates

$$\begin{pmatrix} \phi_{n,A}(x, q_y) \\ \phi_{n,B}(x, q_y) \end{pmatrix} \propto e^{iK_{\xi,x}x}\begin{pmatrix} \pm\phi_{|n|-1}(x, q_y) \\ \phi_{|n|}(x, q_y) \end{pmatrix}, \tag{9}$$

where $\phi_n(x, q_y) = e^{-(x+\xi\ell_B^2 q_y)^2/2\ell_B^2} H_n(x + \ell_B^2 q_y)$ is the $n$-th Landau level state in the Landau gauge, with $H_n(x)$ the $n$-th Hermite polynomial. We finally note that to account for the tilt in the $|n| \geq 1$ levels (see Fig. 1b), one can expand the function $h$ to next order (in $q$ and $x$), finding $\varepsilon_n = \mathrm{sgn}(n) t \sqrt{\tau |n| (1 - \xi q_y d)}$.

We can now see *a posteriori* that our approximation is good as a long as the hopping $t_1(x)$ does not change significantly in a distance $\ell_B$—the width of the Landau level states. The variation is giving by $[t_1(x + \ell_B) - t_1(x)]/t = \sqrt{\tau/2}$.

## B    Numerical simulations

The symbolic hopping (Hop) terms in Eqs. (4) and (5) read

$$
\begin{aligned}
i\hbar\partial_t \psi_A(\mathbf{r} - \mathbf{R}_1) &= -t[\psi_B(\mathbf{r} - \mathbf{R}_1 + \mathbf{R}_2) + \psi_B(\mathbf{r} - \mathbf{R}_1 + \mathbf{R}_3)] - t_1(x)\psi_B(\mathbf{r}) + \dots, \\
i\hbar\partial_t \psi_B(\mathbf{r}) &= -t[\psi_A(\mathbf{r} - \mathbf{R}_3) + \psi_A(\mathbf{r} - \mathbf{R}_2)] - t_1(x)\psi_A(\mathbf{r} - \mathbf{R}_1) + \dots,
\end{aligned}
\tag{10}
$$

where the ellipsis refer to all other terms present in Eqs. (4) and (5). The vector $\mathbf{r}$ labels all the discrete positions of $B$ sites.

We carry out the numerical analysis by solving the stochastic Gross-Pitaevskii equations using the open-source package XMDS2 [61]. For the results shown in Figs. 2 and 4, we first run a single mean-field trajectory ($W_{A,B} = 0$) until a stationary state is reached, typically for a time between $10^3$ and $5 \times 10^3$ in units of the inverse decay rate $\gamma^{-1}$. Only then, we turn on the Gaussian noise and follow up 100 stochastic trajectories for a time $120\gamma^{-1}$. The time Fourier transform in the spectral densities is obtained from the final $\Delta t = 100\gamma^{-1}$ time lapse.

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
