# Peer review of "Polariton condensation into vortex states in the synthetic magnetic field of a strained honeycomb lattice"

_SciPost Physics, doi:SciPost Phys. 12, 068 (2022)_

## Round 1 · Referee Report · Guillaume Malpuech (Referee 1) · 2021-7-2

Report

In this manuscript, the authors consider theoretically Bose Einstein condensation in a strained honeycomb lattice. The strained honeycomb lattice was previously considered in several previous works. The effect of the strain can be modelled as an emergent magnetic field acting on the lattice particles as if they were charged. The sign of the field is opposite in both valleys. The resulting eigen states are well described as artificial Landau levels. On top of this, the authors use a condensation model supposed to describe polariton condensation. They show that by using an appropriate pump profile in real space (pumping only the B-atoms), condensation preferentially occurs in the n=0 Landau levels because these states have the best spatial overlap with the pump. The intrinsic angular momentum carried by these states results in the formation of quantum vortices. The authors show some regimes where condensation occurs in a single valley, which results in the formation of a vortex lattice with all vortices having the same sign.
I found the manuscript very interesting, and I certainly recommend publication. I have a few comments and questions below.

1 A key aspect is that the n=0 Landau levels are B polarized. May be it was discussed in details in previous publications on strained honeycomb lattices, but it is not evident why it is so. It is not completely evident why the A-B symmetry is broken.
2 The n=0 Landau level is favored because it has the best spatial overlap with the pump. Other important aspects which are not really taken into account are the state life-time and energy relaxation. The life time might vary because of the change of the exciton-photon fraction, but also because of the profile of the wave function. Antisymmetric states are more localized in pillars and show longer lifetime than symmetric ones which are more in contact with the edges of the pillar. This type of effect is beyond the tight binding limit used, but it could even more favor condensation in this n=0 state. May be the authors could discuss a little bit more these questions.

3 The next point is about the symmetry breaking effect, namely the choice of a specific valley. This is the key point of the manuscript, and the explanation given is a bit elusive. “On the contrary, interactions play instead a key role in the case of wider incoherent pump profiles, where they make condensation in a single Dirac valley—as opposed to in both valleys simultaneously—more likely. In standard BECs at thermal equilibrium, it is well known [46] that repulsive interactions tend to suppress fragmentation of a BEC. Here, a similar mechanism hinders the condensate from occupying both valleys at the same time.”
This is not clear at all for me. Repulsive interactions favor the homogeneity of the condensate density in real space. If we consider a Kibble Zurek mechanism for the condensation, domains of homogeneous phase should show up when the condensate form. These domains are separated by domain walls associated with a finite kinetic energy cost. These domain walls can then disappear or not. The domain size depends on the healing length and therefore on interactions. In order to have all vortices having the same size, one needs to form domains larger in size than the pump. Another aspect is that interactions lead to the formation of vortices which are topological defect. So they can annihilate, but they cannot scatter in the other valley, which might be the case in the linear regime I guess ? In general I would suggest that the authors, could, may be, elaborate a bit more on these questions. I would suggest to cite two works which might be relevant:
arXiv:2103.06009 where condensation (at the gamma point) in a polariton graphene lattice (but without strain) is considered. This leads to the formation of vortex anti-vortex pairs following the Kibble Zurek scenario.
Nat. Com. 9, 3991, 2018. which considers a staggered honeycomb lattice. Here, it is shown that the filling of the lattice by a BEC makes that Valley excitations are quantum vortices. This is suppressing inter-valley scattering because of the topological protection associated with of the quantum vortex winding number.

4 In the abstract and outlook, it is said that this work paves the way toward the fractional quantum Hall regime. It sounds nice. It is an attractive goal for many of us, and this paper can be useful in that perspective. On the other hand, I personally don’t so clearly see the paving of the way. Here we are approaching the integer quantum hall effect, in a way which makes it extremely difficult to observe experimentally. Going toward the fractional requires much more, and it is not even clear for me if it is possible at all. So except if some clear arguments, explanations can be brought, I would suggest to tone down a bit the fact that this is a goal easy to achieve on a short time scale.

---

## Round 2 · Referee Report · Guillaume Malpuech (Referee 1) · 2021-11-9

Report

I would like first to apologize for the delay in my reply which is partly due to technical problems between scipost emails and the french university network Renater...

I thank the authors for their reply and I think the manuscript can be published without further changes.

I have still a few remarks but they don’t request actions from the authors.

Regarding the question 2 about mode selection, the overlap with the pump is reduced by a factor 2 for states being equally on A and B with respect to states B polarized. This type of factor 2 can easily be reached by the other mechanisms I was mentioning. So it is our theoretician right not to take them into account, but in a real specific case, they would need to be evaluated.

Regarding point 3, I now better understand the point. On the other hand, I would not expect the m=1 and m=-1 state being spatially at the same position. I understand also that the authors really want to discuss an equilibrium steady state and not the consequence of some transient effect and it is perfectly fine to justify that taking these effects into account is beyond the scope of the present work. On the other hand, this transient necessarily occurs when the condensate forms (also in the numerical simulation). Traces of this transient may indeed survive in the steady state. This is especially true in this lattice configuration where vortices cannot move easily once formed.

Best regards,
Guillaume MALPUECH

---

## Round 2 · Referee Report · Anonymous (Referee 2) · 2021-12-19

Strengths

  1. interesting effect that combines interaction and strain-induced synthetic field to produce an effectively rotating polariton condensate.
  2. The manuscript is well written and contains clear results. It is likely to stimulate experiments.

Weaknesses

interaction effects are small and cannot bring the system to the even more interesting fractional Hall regime

Report

The manuscript combines a series of insights to propose an alternative system for observing strongly rotating bosons. It is clear in its presentation and objectives, and I believe it could be published with minor edits. 1. It would be helpful to have an explanation of the pump-noise correlation below eq 3. Also, the sign of gamma in eq 3 and in the pump-noise expression is opposite. Is one of them a typo? 2. I couldn't find a discussion of the experimentally achievable interaction strength. I suggest such a discussion be added. 3. I suggest adding citations to other works producing a synthetic rotation in atomic systems: https://arxiv.org/abs/1901.03705 https://journals.aps.org/prl/abstract/10.1103/PhysRevLett.112.043001 https://iopscience.iop.org/article/10.1088/0034-4885/77/12/126401 and so on. I would encourage the authors to cite more broadly also key works on topological polaritonics.

---

## Round 2 · Author Response

Dear Editor,

Please find submitted a new version of our manuscript titled “Polariton condensation into vortex states in the synthetic magnetic field of a strained honeycomb lattice”, for publication in SciPost Physics. We have effected anew a series of amendments in the main text.

Reply to the Referee:

We thank the reviewer for his comments and his positive assessment. We have addressed his comments in the revised version of the manuscript.

  1. The choice in the direction of the strain is what breaks the symmetry between A and B sites and thus determines the A or B sublattice polarization of the n=0 Landau level (see [29-32] and Appendix A). For positive (negative) strain parameter, $\tau>0$ ($\tau<0$), the $n=0$ wavefunction is localized in the B (A) sublattice. This is mentioned in the paragraph preceding Eq.(3), ``... completely localized in the B sublattice (it would be in the $A$ sublattice if $\tau$ was negative)''. In the revised text, we have added reference to a few works where the interested reader can find a discussion of this fact. A summary of the analytical derivation is available in Appendix A.

  2. This is an interesting point. At the end of Sec.2.1, we have added a sentence explaining why we expect that the mode selection mechanism based on spatial overlap with the pump should largely exceed all other mode selection mechanisms.

  3. Indeed, as the referee points out, what we have in mind is the homogeneity of the condensate density in real space. Among the available states in which condensation can occur (the degenerate $n=0$ Landau level states), it is favourable to choose a linear combination in a single valley to reduce the interaction energy. When only positive or negative angular momenta states are available, like in rotating cold atoms, the energy is minimized forming a triangular array of vortices or antivortices, which the result of Abrikosov. But what happens if both positive and negative angular momenta are available like in our case? A linear combination of states with both chiralities would entail strong inhomogeneities in real space due to the destructive interference. To give an analytical example, let us imagine for a moment that we are dealing with a continuous space like in rotating BECs. Forming an equally-weighted linear combination of $m=1$ and $m=-1$ single-particle states, would increase, compared to a state with just $m=1$ or $-1$, the interaction energy ($\propto \int |\psi(r)|^4 d^2 r$) by a factor of $\left(\int\limits_0^{2\pi}4\cos^4(\phi)d\phi\right)/\left(\int\limits_0^{2\pi}d\phi\right) = 3/2$. Thus a single orientation is favoured.

We have modified our previous sentence, highlighted by the referee, by ``On the contrary, interactions play instead a key role in the case of wider incoherent pump profiles, where they make condensation in a single Dirac valley---as opposed to in both valleys simultaneously---more likely. In standard BECs at thermal equilibrium, the interaction energy is minimized by choosing among the single-particle ground states the one which has the most homogenous density in real space. Here, a similar mechanism hinders the condensate from occupying both valleys at the same time, which would entail large density inhomogeneities due to destructive interference of states with opposite angular momenta.''

Regarding the the two suggested papers, they are clearly relevant and we now cite them. We have now clarified in the manuscript, though, that we are not dealing with transient phenomena like Kibble-Zurek physics, which is beyond the scope of our work.

  1. We have refined the tuning of the sentences in the outlook modifying the last paragraph: ``When applied to strongly-interacting photonic lattices...''.

In view of the above modifications and qualifications, we would be grateful if you could consider our manuscript for publication in SciPost Physics.

Yours sincerely,

C. Lledó, I. Carusotto, M. H. Szymaǹska

---

## Round 2 · List of Changes

1) Added references [44, 45, 47, 50].

2) In the last paragraph of the Introduction, we changed the sentence "...as used in the recent works [27, 40–43]..." for "...as used in the recent experiments [27,40–43] and theoretical works [31,44,45]..."

3) In the paragraph following Eq. (3), we changed "...We will take advantage of this to induce polariton condensation into the n = 0 Landau level." for an extended version "...We will take advantage of this to induce polariton condensation into the n = 0 Landau level [47]: given the full sublattice polarization of this state, we expect that the mode selection mechanism based on spatial overlap with the pumped sublattice will largely exceed other mechanisms that may be active in polariton systems, such as energy relaxation, the exciton-photon fraction of modes, and the amount of localization of the wave function at the pillars."

4) We have corrected a typo at the beginning of Sec. 2.3 : "$P_B >~ \gamma$" has been replaced for $P_B \gtrsim \gamma$.

5) At the end of Sec. 2.2 we have included the new paragraph "The results that we are going to show in the following of the work refer to the long-time limit of the system evolution. A study of the additional features that may occur during the switch-on transient, such as Kibble-Zurek vortex nucleation phenomena [45] go beyond the scope of this work."

6) We have have erased the words "...the y-reflection symmetry is explicitly broken..." from the first paragraph of the Discussion section, as it seemed to obscure the proposal.

7) We have rewritten the last paragraph of the Outlook.

8) We have added a "Data and Code Availability" section before the Acknowledgements.

---

## Round 3 · List of Changes

1) We added references 4-7 to the introduction.
2) Below Eq.(5), we added the following sentence:
"Polaritons in micropillars have an on-site interaction energy smaller than the linewidth, $U/\hbar\gamma < 1$; in our results below we use modest ratios perfectly within the reach of current experiments (see e.g. Ref. [52])."
3) We added the footnote in page 5 to explain the origin of the form of the pump and of the noise correlations.
4) We submitted to the UCL data repository the videos we had accompanying Figures 3a and 3b. Reference 53 points to them.

---

## Editorial Decision

published